# Association Between Renal Cell Cancer and Chronic Kidney Disease: An Update on a Never-Healing Wound

**DOI:** 10.3390/biomedicines13112638

**Published:** 2025-10-28

**Authors:** Ilias Giannakodimos, Aris Kaltsas, Andreas Koumenis, Evangelia Mitakidi, Konstantinos Adamos, Dimitrios Deligiannis, Marios Stavropoulos, Zisis Kratiras, Michael Chrisofos

**Affiliations:** 1Third Department of Urology, Attikon University Hospital, School of Medicine, National and Kapodistrian University of Athens, 12462 Athens, Greece; ares-kaltsas@hotmail.com (A.K.); a_koumeni@icloud.com (A.K.); constantinos.adamos@gmail.com (K.A.); d.delijohn@yahoo.gr (D.D.); stamarios@yahoo.gr (M.S.); zkratiras@gmail.com (Z.K.); mchrysof@med.uoa.gr (M.C.); 2Departement of Anesthesiology, General Hospital of KAT, 14561 Athens, Greece; evangeliamitakidi@gmail.com

**Keywords:** renal cell cancer, renal disease, chronic kidney disease, renal surgery, risk factors

## Abstract

The relationship between chronic kidney disease (CKD) and renal cell carcinoma (RCC) is both bidirectional and multifactorial. Several risk factors, including hypertension, diabetes mellitus, obesity, and smoking, have been associated with an increased risk for the development of CKD and RCC. CKD may predispose individuals to RCC through various mechanisms, including renal cystic diseases or induced oxidative stress effects. Conversely, RCC can induce CKD through the direct effects of the tumor, after surgeries for the management of the tumor (either partial or radical nephrectomy), and through perioperative acute kidney injury. Furthermore, medical interventions, including immunotherapy or targeted therapies, may precipitate acute kidney injury, potentially leading to the development of CKD. The expression of several genes in renal tissues has been related to the remodeling of kidneys during end-stage kidney disease and with an increased risk of the development of preneoplastic lesions and tumors. The aim of this review is to update the knowledge of these relationships, highlight the pathophysiologic mechanisms, and identify the genes and molecular expressions involved in this pathway.

## 1. Introduction

Renal cell carcinoma (RCC) is a significant urologic malignancy worldwide, with an estimated ~434,800 new cases in 2022 [1]. In recent decades the incidence of RCC has advanced due to better imaging modalities and earlier diagnosis [2]. The relationship between RCC and renal dysfunction is multifactorial: chronic kidney disease (CKD) can increase RCC risk, and conversely, RCC and its treatment, can impair renal function [3]. This association between RCC development and the occurrence of CKD is demonstrated in Figure 1. As a result, urologists have to deal with patients with renal disease, since both the development of RCC itself and its management can lead to acute kidney injury (AKI) and eventually to chronic kidney disease (CKD) [4]. Additionally, several other risk factors, such as hypertension, diabetes, obesity, and smoking, may predispose and contribute to the development of both CKD and RCC [3]. Interestingly, in patients with underlying renal disease—from mild CKD to end-stage renal disease—RCC incidence is elevated, whereas among RCC patients, renal impairment is common due to baseline comorbidities, paraneoplastic glomerular injury, or the loss of nephron mass after nephrectomy [3]. The aim of this review was to update the knowledge on the bidirectional linkage between RCC and renal disease or CKD development by highlighting the epidemiology, pathophysiological mechanisms, risk factors, and treatment modalities of this relationship. Also, the main purpose of this review was to describe the proper management of RCC cases in ESKD patients and provide future directions for this issue.

## 2. Literature Search

We reviewed the current literature from the inception of this current review until August 2025. For our scope, we used the “PubMed” database and we included only studies written in English. The search was based on the following search terms: “Renal cell cancer”, “renal disease”, “End-stage renal disease”, “chronic kidney disease”, and “transplant recipients” and we retrieved the results of our search on the topics of epidemiology, pathophysiologic mechanisms, risk factors, and treatment. Also, the references of the research articles were scrutinized for relevant studies.

Only studies that concerned human adults (≥18 years) with CKD, ESKD, and/or diagnosed RCC were included. Retrospective or prospective studies, epidemiological research, systematic reviews, and/or meta-analyses that concerned CKD or ESKD patients (dialysis or transplant) or patients with RCC were included in our study. RCC cases were defined by histological diagnosis or validated cancer registry coding. Studies with interventions related to RCC management (partial/radical nephrectomy, ablation, systemic therapies including immunotherapy/targeted agents) and their renal effects were also included. Finally, studies that referred to pediatric populations (<18 years), do not define CKD or RCC using standard clinical or histologic criteria, case reports, and non-English studies were excluded.

## 3. RCC in Patients with Renal Disease

### 3.1. Epidemiology of RCC in Patients with Renal Disease

Several studies have reported an increased incidence of RCC in patients with CKD, ranging from no risk to a 100-fold increased risk [5,6]. However, recently published studies suggest that even moderate CKD confers a higher risk of RCC. More specifically, in a large Korean cohort study of 9,809,317 adults, during a median follow-up period of 7.3 years, 10,634 kidney cancers were identified. A reduced glomerular filtration rate (GFR) was associated with increased kidney cancer; stage 3 CKD (eGFR 30–59) was linked to a ~22% higher RCC risk, and stage 4 CKD (eGFR < 30) to an ~18% higher risk (relative to eGFR 60–89) [7]. Interestingly, the presence of proteinuria further amplified this risk, suggesting a synergistic effect of impaired GFR and kidney damage [7]. Furthermore, other epidemiologic studies have reported a 2–3-fold higher risk of RCC in CKD patients [8]. For instance, a US case–control analysis, comparing 3136 renal cell carcinoma cases diagnosed between 1998 and 2008 with 31,031 individually matched controls, showed that patients with CKD presented with a three-fold higher RCC risk [8]. As a result, these findings strongly indicate that CKD, even at earlier stages, comprised a strong independent risk factor for RCC development.

Additionally, patients with end-stage renal disease (ESRD) also present with a markedly elevated RCC incidence. A recent nationwide study from Sweden, including 9299 patients with RCC identified from 2005 until 2014 and 93,895 matched controls, reported a 4.5-fold higher risk of RCC in ESRD patients compared with the general population [9]. Notably, an association between the duration of renal replacement therapy and RCC risk development was reported; ESRD patients on long-term dialysis (over 9 years) presented with a ~10-fold increase in RCC odds [9]. More specifically, RCC in the ESRD setting often involves distinct histological subtypes associated with ACKD. The two most frequent RCC subtypes in ESRD are ACKD-associated RCC and clear cell papillary RCC, which are recognized as distinct entities in the WHO classification and occur at much higher frequency in dialysis patients than in sporadic RCC [10]. Of note, ESRD-associated RCCs are also more often multifocal and bilateral. Despite the increased incidence, these tumors tend to be detected at a younger age and earlier stage in dialysis patients, likely owing to imaging surveillance of longstanding ESRD [9]. On the contrary, patients with AKI are at a higher risk of developing papillary RCC.

Epidemiological studies indicate that the incidence of RCC in KTRs is ~0.5–0.7%, representing a 6–7-fold higher risk than in age-matched general populations [11]. Of note, the vast majority of these RCC cases arise not in an allograft kidney but in the patient’s native kidneys [11]. Interestingly, RCC in native kidneys post transplant is often asymptomatic and detected on routine imaging; in contrast, de novo cancers in a transplanted kidney allograft are rare.

### 3.2. Risk Factors for the Development of RCC in Patients with Renal Disease

Finally, the association of RCC development in transplant recipient patients is also considered unique. Although kidney transplant recipients (KTRs) carry an elevated cancer risk in general, due to immunosuppression and oncogenic viruses, RCC constitutes the most common post-transplant malignancy involving the urinary system. Several factors have been implicated for the development of RCC in patients with renal disease. The majority of studies report a 3–7-fold increased risk of RCC development in CKD patients compared with the general population [12,13,14]. It seems that kidney disease stimulates a kidney-specific carcinogenic effect that promotes RCC development [15,16]. This oncogenic environment in renal disease patients is justified by the different characteristics reported between sporadic and CKD-related RCCs and is considered dose-dependent, since a lower eGFR is related to a higher risk of RCC [17]. As already mentioned, renal injury in certain renal areas may cause the development of specific types of kidney cancers [3]. More specifically, cc-RCC is usually developed in the injured areas of proximal tubules (S1/S2 segment) [18]. In the same renal area, cortical damage is caused by the metabolic load of diabetes mellitus and obesity, increasing the risk for the development of RCC [19]. On the contrary, renal damage located in the outer medulla of proximal tubule (S3 segment) is associated with an increased risk of papillary adenoma, that may eventually transform into papillary RCC [20]. Finally, the vascular injury of the kidney may be the intrinsic cause of kidney disease and is related to the development of papillary RCC rather than cc-RCC [21].

Furthermore, patients with ESKD are related to an increased risk of RCC-associated ACKD [3]. The relationship of ACKD to the progression from initial areas of renal lesions to adenoma and finally, to RCC has been well established [22,23]. In ESKD patients, male sex, age, tuberous sclerosis, focal segmental glomerulosclerosis, and ACKD are considered to be independent risk factors for the development of RCC [24]. However, compared with patients with sporadic tumors, RCC-EKSD patients are found with lower stage and grade tumors, presenting with better cancer-specific survival [3]. Interestingly, although albuminuria comprised a known sign of renal damage and progression to CKD, it may also reflect microvascular damage or affected glomerular permeability caused by tumor products [25]. Increased levels of urine albumin are related to advanced tumor stages and higher tumor burden, which are significant factors for cancer-related death [26]. As a result, albuminuria may reflect a worse systematic or tumor burden and is related to an increased risk of cancer death in RCC patients, associated with worse survival and more aggressive disease [26]. The risk of RCC in ESKD patients increases in a dose-dependent manner with time on dialysis, whereas it decreases after kidney transplantation [25,26]. According to a US registry of 116,208 kidney recipients from 1987 to 2010, an increased RCC risk is observed in transplant recipients, reaching a nadir after 2.5 years and steadily rising in the following post-transplant years [25,26]. Risk factors for RCC in KTRs include the same factors as in dialysis patients; long pre-transplant dialysis duration, ACKD, and immunosuppressive treatment [3]. Due to the elevated risk of RCC, the lifelong surveillance of native kidneys is recommended in transplant recipients with risk factors, especially in patients with a prolonged dialysis history before transplant [11]. The risk factors related to the development of RCC in patients with renal disease are presented in Table 1.

### 3.3. Pathophysiologic Mechanisms of RCC Development in Patients with Renal Disease

#### 3.3.1. The Role of Chronic Inflammation, Oxidative Stress, and Immune Deficiency

ESKD is marked by persistent inflammation, while uremia, a hallmark of ESKD, places stress on multiple tissues, thereby also triggering a systemic inflammatory response [27]. Individuals with ESKD often present with increased levels of circulating pro-inflammatory cytokines such as interleukin-1β, interleukin-6, and tumor necrosis factor-α, which together establish a chronic, low-grade inflammatory state that promotes vascular and tissue damage [27]. In patients receiving hemodialysis, the procedure itself can intensify this inflammation by stimulating immune cells through interaction with the dialysis membrane [28]. This ongoing inflammatory condition can activate downstream molecular pathways, including the c-Jun pathway, which is thought to raise the risk of developing RCC [28]. Along with inflammation, oxidative stress also plays a significant role in the development of renal cancer in ESKD patients [27]. More specifically, there exists an imbalance between the generation of reactive oxygen species (ROS) and the body’s antioxidant defenses [27]. Of note, both uremia and the dialysis process contribute to increased ROS production and reduced antioxidant capacity [28]. In addition, oxidative stress triggers DNA damage, including strand breaks and base modifications, which can elevate mutation rates [28]. Compared with individuals with normal kidney function, those undergoing dialysis exhibit elevated levels of oxidative stress markers within renal tissue [10]. The surplus of reactive oxygen species (ROS) causes progressive harm to key cellular macromolecules (DNA, proteins, and lipids) fostering mutagenesis and genomic instability [29]. Finally, oxidative stress acts in combination with chronic inflammation by stimulating signaling pathways that promote cell proliferation, suppress programmed cell death, and support angiogenesis, which are considered crucial for tumor advancement [30].

ESKD patients experience marked dysfunction of the immune system, which may elevate their susceptibility to cancer [31]. The uremic condition characteristic of ESKD contributes to immunosuppression by impairing the activity of various immune cell types [31]. In patients undergoing hemodialysis, the repeated exposure of blood to the dialysis membrane can lead to abnormal immune cell activation, resulting over time in a form of acquired immunodeficiency, creating a biological environment that favors tumor development [28]. This weakened immune surveillance means that ESKD patients are less capable of recognizing and eliminating pre-malignant cells, allowing them to proliferate [28]. Although the specific mechanisms for how the immune system during ESKD is related to RCC development are not fully outlined in current research, several mechanisms have been reported, such as the reduced expression of major histocompatibility complex (MHC) class I molecules, the increased expression of immune checkpoint proteins like programmed death ligand 1 (PD-L1), and the secretion of immunosuppressive cytokines.

#### 3.3.2. ACKD as a Precursor to Malignancy

ACKD is a frequent complication affecting roughly 20% of ESKD individuals, especially those receiving long-term dialysis therapy [32]. This condition is marked by the development of numerous cysts within the kidneys and its incidence increases with the duration of dialysis treatment [32]. ACKD is a recognized risk factor for RCC, particularly a distinct subtype known as acquired cystic kidney disease-associated RCC (ACKD-RCC) [32]. Another RCC variant, the clear cell papillary tumor (CCPT), also shows elevated incidence in ESKD kidneys and is often linked to the presence of ACKD [32]. In addition, newly described or provisional tumor types—such as eosinophilic solid and cystic (ESC) RCC and low-grade oncocytic tumors—are found more commonly in patients with ESKD [32]. The cystic transformation of kidneys in ESKD appears to foster a microenvironment that favors the emergence of these specific RCC subtypes, indicating a potential causal relationship between cyst formation and tumorigenesis [28]. The development of ACKD involves the proliferation of the epithelial cells lining the cysts [28]. Research has demonstrated an increased expression of hepatocyte growth factor (HGF) and its receptor c-Met in ACKD kidneys with RCC, as well as within hyperplastic cysts themselves [10]. Additionally, elevated levels of anti-apoptotic proteins such as Bcl-2 in the cystic epithelium indicate heightened proliferative activity within these cysts [28]. It is postulated that these hyperplastic cysts could represent precursor lesions for both ACKD-RCC and CCPT [10].

#### 3.3.3. Epigenetic Alterations and Molecular Pathways in ESKD-Associated RCC

Renal cancers that develop in ESKD patients display a diverse array of genetic mutations and chromosomal changes that are uniquely associated with the ESKD environment [33]. More specifically, cc-RCC arising in ESKD patients frequently carry mutations in the same key driver genes as sporadic cc-RCCs [33]. However, studies have linked longer durations of dialysis treatment with the emergence of a rare mutational pattern known as SBS23 [33]. In addition, Johnson et al. revealed that all cases of acquired cystic disease-associated RCC (ACD-RCC) in ESKD patients exhibit an amplification of chromosome 16q [33]. In addition to genetic mutations, epigenetic alterations also play a critical role in renal carcinogenesis in the ESKD population. One such alteration—hypermethylation—can silence tumor suppressor genes and has been implicated in the development of RCC [32].

Overall, the molecular landscape of renal cancer in ESKD is complex, reflecting both shared and distinct features when compared to sporadic RCC [34]. Multiple critical molecular signaling pathways involved in regulating cell growth, survival, angiogenesis, and inflammation are commonly disrupted in renal cancers associated with ESKD [35]. Among these, the hypoxia-inducible factor (HIF) and mammalian target of rapamycin (mTOR) pathways are particularly significant in the development of cc-RCC [36]. The hepatocyte growth factor (HGF)-c-Met signaling axis is also implicated in cyst formation and its progression to RCC within the ESKD setting [10]. In ESKD, the chronic activation of RAS, characterized by elevated levels of intrarenal angiotensin II and the increased expression of the AT1 receptor, contributes to kidney injury through mechanisms like tubulointerstitial fibrosis. RAS may also facilitate cancer development by enhancing angiogenesis, cellular proliferation, and metastasis, though it remains unclear whether these oncogenic effects are a cause or consequence of the disease process [10]. Other key pathways include Wnt/β-catenin, which governs cell proliferation and differentiation and is frequently dysregulated in RCC [35]. Wnt/β-catenin activation is a well-established oncogenic driver. More specifically, the accumulation of β-catenin in the nucleus leads to the transcription of target genes, such as c-Myc and cyclin D1, which promote uncontrolled cell growth and inhibit apoptosis. The chronic inflammatory state in CKD can activate this pathway, creating a pro-carcinogenic environment. The transforming growth factor-β (TGF-β) pathway, essential for normal glomerular and tubular function, exhibits dual roles: it can act as a tumor suppressor in early RCC stages but later promote tumor invasion and metastasis, particularly when aberrantly activated in ESKD-associated renal fibrosis [10]. Additionally, the nuclear factor erythroid 2-related factor 2 (Nrf2) pathway—a central regulator of antioxidant responses—is often overexpressed in RCC and is under investigation as a therapeutic target [37]. In the presence of oxidative stress, Nrf2 is released, translocates to the nucleus, and binds to antioxidant response elements (AREs), promoting the transcription of antioxidant and detoxification genes. While Nrf2 activation can initially suppress cancer development by protecting against DNA damage, a sustained and chronic activation can make cancer cells resistant to chemotherapy and radiotherapy. The interplay between the Wnt/β-catenin and Nrf2 pathways is a key area of research. Of note, several studies suggest that Wnt/β-catenin can modulate Nrf2 activity, and vice versa, which can contribute to the complex relationship between CKD and cancer. The pathological conditions characteristic of ESKD, including chronic inflammation and oxidative stress driven by uremia, directly influence the activity of these signaling networks in renal cells, thereby contributing to tumor development and progression [27]. Molecular pathways in ESKD-associated RCC are presented in Figure 2.

## 4. Renal Disease in Patients with RCC

### 4.1. Epidemiology of Renal Disease in Patients with RCC

Many patients with RCC already have reduced renal function at presentation, and thus, the surgical management of RCC can further compromise kidney function. The loss of renal mass from tumor nephrectomy is a well-recognized cause of CKD in RCC survivors. More specifically, as far as small renal tumors are concerned, radical nephrectomy carries a high risk of postoperative CKD, whereas partial nephrectomy greatly mitigates this risk. According to a retrospective cohort study of 10,886 patients who underwent partial or radical nephrectomy between 1991 and 2002, approximately 65% of patients undergoing radical nephrectomy and only 20% of patients that underwent partial nephrectomy developed new-onset CKD (eGFR < 60), respectively [38,39]. In other words, nephron-sparing surgery reduced the incidence of stage 3 CKD by about two-thirds. In addition, the severity of CKD is also worse with radical surgery; post-surgical eGFR was estimated at <45 (Stage 3b CKD) in 36% of radical nephrectomy patients, compared with 5% after partial nephrectomy [38,39]. Consequently, current guidelines emphasize kidney-sparing techniques for localized RCC whenever oncologically appropriate, especially in patients with pre-existing renal dysfunction or a solitary kidney [40]. Despite these efforts, a subset of RCC patients will finally progress to end-stage renal disease after surgery. This is emphasized by a national Swedish registry study of 16,220 patients with RCC and 162,199 controls across multiple national registers between 2005 and 2020 that showed that among >16,000 RCC patients, the 5-year cumulative incidence of ESRD after RCC diagnosis was 2.4%, significantly higher than the ~0.4% in matched controls [41]. Age, chronic kidney disease, higher T-stage, and radical nephrectomy were significant risk factors for ESRD within 1 year of surgery [41]. In the same cohort, concerning patients that underwent radical nephrectomy, older age, baseline CKD, and higher tumor stage comprised independent risk factors for post-nephrectomy ESRD [41]. These findings underscore that renal functional outcomes should be considered of significant importance in RCC treatment decisions. Multidisciplinary follow-up of RCC survivors should include monitoring for CKD development and the management of cardiovascular risk factors associated with CKD.

Although uncommon, RCC can precipitate paraneoplastic glomerular diseases that cause renal impairment. A 2023 review identified 35 reported cases of RCC-associated paraneoplastic glomerulopathy in the literature [42]. The majority of patients were older males (median age > 60), and membranous nephropathy was the single most common pathological subtype (34% of cases) [42]. In most instances, the glomerular disease was diagnosed either concurrent with the RCC or preceding the cancer diagnosis (often manifesting as proteinuria/nephrotic syndrome that led to the tumor’s discovery) [42].

### 4.2. Risk Factors for the Development of Renal Disease in RCC Patients

It is increasingly recognized that many patients with RCC harbor coexistent chronic kidney damage from other causes, owing to shared risk factors and the demographic profile of RCC. These risk factors include various demographic characteristics, such as age, gender, ethnicity, lifestyle habits, such as smoking and nutrition, genetic factors, and other comorbidities, such as diabetes mellitus and hypertension [43]. Cross-sectional studies show that a substantial fraction of RCC patients meet the criteria for CKD at the time of diagnosis. More specifically, an analysis of a retrospective cohort study of 662 patients that underwent partial or radical nephrectomy between 1989 and 2005 found that approximately 26% of patients had pre-existing CKD (eGFR < 60) before nephrectomy for a small renal tumor [44]. Similarly, a narrative review noted that 22–36% of RCC patients have CKD prior to surgery in various series [3]. Interestingly, tumor diameter is related to preoperative eGFR and the postoperative development of CKD [45]. In another retrospective study that included 1569 patients undergoing surgery for renal cortical tumors, several tumor characteristics, such as tumor size, stage, and histology were associated with worse CKD risk stratification [46]. This is due to a decrease in functional nephrons of the kidney caused by immediate pressure from the tumor [46]. Finally, worse baseline renal function is correlated with an elevated risk of a papillary RCC tumor [47].

Furthermore, postoperative renal function is associated with preoperative tumor burden, since larger tumors are associated with worsening kidney function, especially in patients undergoing partial nephrectomy [45,48]. On the contrary, another retrospective study that analyzed data from 944 kidney cancer patients managed with radical nephrectomy and 242 living kidney donors who underwent surgery showed that in patients that underwent radical nephrectomy, larger tumors were associated with worse preoperative renal function, but a lower risk of worsening renal function postoperatively, due to the decreasing portion of normal parenchyma lost with these tumors [49]. Of note, hyperfiltration injury to the remaining renal parenchyma after partial or radical nephrectomy might predispose patients to worse long-term renal function [43].

The overlap of risk factors likely contributes to this intersection: epidemiologic data confirm that conditions like obesity, smoking, and metabolic syndrome are associated with higher risks of both RCC and CKD [3]. More specifically, in comparison with healthy donors, RCC patients were found to be older and obese with vascular disease, diabetes, or hypertension [3]. The majority of these patients also have unrecognized CKD or proteinuria before surgical intervention [3]. Another study showed that independent factors, such as interstitial fibrosis, tubular atrophy, arteriolar sclerosis, female sex, and pre-existing renal disease, were associated with worse renal function after surgery [3]. Pre-existing comorbidities, such as older age, diabetes, hypertension, and tobacco abuse might decrease renal function after nephrectomy and promote renal insufficiency due to their long-term effects on the remaining kidney [43]. Additionally, inherited or congenital kidney diseases can predispose to RCC; for instance, patients with polycystic kidney disease (ADPKD) or tuberous sclerosis complex have higher incidences of renal tumors, though these syndromic RCC cases form a small subset of overall RCC burden. Coexisting renal pathologies (such as diabetic glomerulosclerosis or hypertensive nephrosclerosis) are often found in the histologic examination of nephrectomy specimens removed for RCC. The risk factors related to the development of renal disease in patients with RCC are presented in Table 1.

### 4.3. Pathophysiologic Mechanisms of Renal Disease Development in RCC Patients

#### 4.3.1. Direct Tumor Effects on Kidney Dysfunction

RCC can negatively impact kidney function through multiple direct mechanisms. As the tumor enlarges, it can invade and destroy adjacent healthy renal parenchyma [3]. This direct infiltration reduces the kidney’s functional capacity by displacing or eliminating nephrons, leading to a measurable decline in renal filtration. In addition to tissue destruction, RCC can exert mechanical pressure on critical renal structures, further impairing function. One significant complication is the development of tumor thrombus within the renal vein, and in more advanced cases, its extension into the inferior vena cava (IVC), which occurs in approximately 4–10% of RCC patients [50]. This venous invasion obstructs renal venous outflow, leading to venous congestion within the kidney and potentially causing acute kidney injury (AKI) or contributing to chronic kidney disease (CKD) [51]. Furthermore, tumor growth or metastasis can cause the mechanical obstruction of the urinary tract. This can present as ureteral blockage, resulting in hydronephrosis which may ultimately lead to AKI if not promptly resolved [52]. Thus, RCC can impair kidney function both through direct structural destruction and through indirect vascular or urinary tract compression.

Another mechanism suggests that cc-RCC is closely linked to the inactivation of the von Hippel–Lindau (VHL) tumor suppressor gene, a critical event that results in the continuous activation of hypoxia-inducible factor (HIF) transcription factors—particularly HIF-1α and HIF-2α—even in normal oxygen conditions [53]. This constitutive HIF activation drives the expression of a variety of genes that promote tumor growth, angiogenesis (including vascular endothelial growth factor [VEGF], platelet-derived growth factor B, and transforming growth factor alpha), and extensive metabolic reprogramming [53]. The resulting metabolic dysregulation is considered advantageous for tumor proliferation and alters both the local renal microenvironment and systemic metabolic balance, which can indirectly impair kidney function [29]. Simultaneously, the pentose phosphate pathway is upregulated to produce NADPH and nucleotide precursors required for rapid cell division [54]. In contrast, the tricarboxylic acid (TCA) cycle and oxidative phosphorylation are often downregulated in cc-RCC. Mutations in TCA cycle enzymes like fumarase and succinate dehydrogenase lead to the buildup of oncometabolites, which further promote tumorigenesis [54]. Additionally, fatty acid synthesis is enhanced, contributing to intracellular lipid accumulation—a hallmark of cc-RCC—while fatty acid β-oxidation is typically suppressed [54]. This hostile biochemical environment places stress on surrounding healthy nephrons, potentially disrupting their function and contributing to localized kidney damage. In this way, the metabolic phenotype of cc-RCC not only drives malignancy but may also compromise overall renal health.

#### 4.3.2. Systemic Tumor Effects on Kidney Dysfunction

Beyond direct tumor effects, RCC can induce kidney dysfunction through various systemic mechanisms, including the release of humoral factors, the induction of widespread inflammation and oxidative stress, and alterations in systemic hemodynamics and electrolyte balance.

#### 4.3.3. Paraneoplastic Syndromes

RCC is frequently associated with a wide spectrum of paraneoplastic syndromes (PNS) which may affect renal function and clinical disorders that arise at sites distant from the primary tumor or its metastases and are not attributable to direct tumor invasion, obstruction, or infection [55]. These syndromes range from general constitutional symptoms like fever and weight loss to more specific metabolic and biochemical abnormalities [55]. A hallmark of many RCC-associated PNS is their reversibility following successful tumor removal [55]. One of the most common paraneoplastic complications in RCC is Humoral Hypercalcemia of Malignancy (HHM), affecting approximately 17% of patients [56]. This disturbance in calcium homeostasis, caused by the ectopic production of the parathyroid hormone-related peptide, can severely affect kidney function: acute hypercalcemia may provoke acute renal failure and tubular necrosis, while chronic hypercalcemia can result in hypercalciuria, nephrocalcinosis, interstitial fibrosis, and tubular atrophy [57]. Another mechanism suggests that hyperparathyroidism in ESKD patients contributes to renal carcinogenesis by promoting unregulated cell proliferation in the kidneys. High levels of parathyroid hormone (PTH) can act as a growth factor, leading to the rapid division of cells and increasing the likelihood of genetic mutations [58]. This hormone also worsens the body’s already-impaired DNA repair mechanisms in uremic patients, allowing damage to accumulate [58]. Furthermore, PTH-induced hypercalcemia can create an inflammatory environment in the kidneys, which is a known risk factor for cancer development. Together, these processes create a pro-carcinogenic environment, making the kidneys more susceptible to tumor formation [58]. Another rare but important group of RCC-associated paraneoplastic conditions is paraneoplastic glomerulonephropathies (PGNs), which typically present as nephrotic syndrome, nephritic syndrome, or unexplained renal impairment [42,59]. The most frequent renal cancer type associated with PGNs is clear cell RCC, which is believed to arise through immune-mediated injury to the glomeruli initiated by tumor antigens [42]. Stauffer’s syndrome is a distinct paraneoplastic hepatic disorder associated with RCC, characterized by a reversible, non-metastatic elevation in liver enzymes—particularly alkaline phosphatase—without evidence of hepatic metastasis or biliary obstruction [60]. It is often accompanied by elevated ESR, thrombocytosis, and hepatosplenomegaly [60]. Interleukin-6 (IL-6) is thought to play a central role in its pathogenesis, and the syndrome may resolve completely after tumor resection [61]. Although Stauffer’s syndrome does not directly harm the kidneys, the associated systemic inflammatory state can indirectly affect renal health by worsening pre-existing conditions or altering drug metabolism and excretion [61]. Secondary (AA) amyloidosis represents a rare but serious systemic complication of RCC, with the kidney being one of the most frequently affected organs, resulting from the chronic inflammation-induced overproduction of the serum amyloid A protein [62]. In addition to these syndromes, RCC may present with other paraneoplastic manifestations—such as hypertension, anemia, polycythemia, neuromyopathies, vasculopathies, and metabolic disturbances—that may not directly impair renal function but can significantly affect overall health and renal physiology [55]. These manifestations can indirectly contribute to kidney stress by increasing metabolic demands, impairing cardiovascular regulation, or complicating therapeutic management. Early recognition and tumor-specific treatment are critical for managing these manifestations and preserving normal renal function.

#### 4.3.4. Systemic Inflammation and Oxidative Stress

Inflammation is increasingly recognized as a key hallmark of advanced RCC, where it plays a central role in tumor progression, metastasis, and resistance to therapy [63]. The tumor microenvironment (TME) of RCC is heavily infiltrated by diverse immune cell populations, which significantly shape both tumor growth and immune evasion mechanisms [64]. Intrinsic alterations within RCC cells further activate inflammatory signaling pathways, driving the release of chemokines and neoantigen expression, and fostering strong immunosuppression [63]. Importantly, this inflammation is not confined to the tumor site; it extends systemically, as demonstrated by elevated levels of circulating inflammatory mediators in RCC patients [55]. Pro-inflammatory cytokines such as interleukin-1 beta (IL-1β), interleukin-6 (IL-6), and tumor necrosis factor-alpha (TNF-α)—primarily secreted by macrophages and other immune cells in the TME—are instrumental in both tumor progression, systemic immune modulation, and renal function [64,65]. Chronic systemic inflammation can damage non-cancerous kidney tissue by disrupting the renal microvasculature and inducing the production of nephrotoxic factors, including reactive oxygen species [30]. Oxidative stress, characterized by an imbalance between ROS production and antioxidant defenses, is a common feature in RCC, mainly due to higher metabolic activity and rich blood supply. Tumor-driven systemic oxidative stress contributes to kidney injury in non-neoplastic tissue by promoting excess ROS generation and depleting antioxidant reserves [66]. Another key mechanism for carcinogenesis is the formation of N-nitroso compounds (NOCs), which are potent carcinogens, since the kidneys of ESKD patients fail to excrete nitrogenous waste, leading to the accumulation of uremic toxins [67]. The high concentration of nitrogenous compounds in the blood promotes their conversion into these cancer-causing agents [67]. This buildup also contributes to chronic inflammation and oxidative stress, which directly damage cellular DNA and impair the body’s ability to repair itself. This continuous assault on kidney cells from multiple fronts significantly increases the risk of renal cancer. This emerging understanding highlights the broader systemic effects of RCC and suggests that both inflammatory and oxidative mechanisms initiated by the tumor can significantly contribute to renal deterioration, emphasizing the importance of systemic disease management in RCC patients.

#### 4.3.5. Hemodynamic Alterations and Electrolyte Imbalances

Hypertension is both a recognized risk factor for the development of renal cell carcinoma and a possible paraneoplastic manifestation directly driven by the tumor itself [55]. Certain renin-secreting renal tumors—such as RCC, Wilms’ tumors, juxtaglomerular cell tumors, and benign renal hemangiopericytomas—are capable of inducing reversible hypertension through excessive renin production [68]. Elevated plasma renin activity has been observed in the renal veins draining tumor-bearing kidneys, along with an increased renin content within the tumor tissue itself. In many cases, blood pressure normalizes following nephrectomy, further implicating the tumor as the source of hypertension [69]. In addition, other tumor-related mechanisms have been proposed to contribute to hemodynamic alteration in kidney. These include vasculitis, polycythemia secondary to elevated erythropoietin, hypercalcemia, renal arteriovenous fistulas, ureteral obstruction, and the ectopic secretion of noradrenaline [70]. Each of these factors can alter vascular resistance, fluid balance, or systemic vascular tone, thereby contributing to elevated blood pressure and thus affecting kidney function. By increasing intraglomerular pressure, it promotes hyperfiltration injury, which over time damages glomerular structures and accelerates the development of hypertensive nephrosclerosis in the non-tumor-bearing renal tissue [70].

As already mentioned, hypercalcemia constitutes a well-established paraneoplastic electrolyte disturbance in RCC with direct nephrotoxic effects [71]. Other electrolyte abnormalities, such as hypokalemia and metabolic acidosis, are more commonly associated with chronic kidney disease and typically arise as secondary consequences of impaired renal function rather than direct tumor secretion [71]. A profound deficiency of active vitamin D, or calcitriol, is common in ESKD patients and removes a critical anti-cancer defense mechanism. Calcitriol normally has powerful antiproliferative effects, meaning it helps prevent the uncontrolled growth of cells [72]. Without it, kidney cells are more prone to becoming cancerous. The deficiency also impairs cellular differentiation, leading to immature, cancer-like cells that can more easily form tumors [72]. This loss of normal gene regulation creates a highly favorable environment for the initiation and progression of cancer [72]. Once RCC-induced kidney dysfunction has initiated, secondary disturbances in electrolyte and acid–base homeostasis frequently develop. Hypokalemia, for instance, may progress into hypokalemic nephropathy, a tubulointerstitial disorder marked by prolonged potassium deficiency [73]. This condition can induce renal vasoconstriction, diminish medullary blood flow, and result in intrarenal ischemia, thereby exacerbating renal injury [73]. Metabolic acidosis is another common complication in advanced CKD, characterized by the accumulation of hydrogen ions due to the kidneys’ reduced ability to excrete daily acid loads which accelerates glomerular filtration rate (GFR) decline [66]. Thus, although not always directly initiated by tumor-secreted factors, these electrolyte and acid–base imbalances represent important downstream effects of RCC-related kidney dysfunction. The molecular pathways in ESKD-associated RCC are presented in Figure 2

## 5. Management of RCC in ESKD Patients

The surgical resection of a tumor, via open, laparoscopic or robotic approaches, constitutes the mainstay of treatment for localized disease and displays favorable outcomes [40]. As already mentioned, partial nephrectomy comprises the preferred surgical option, since it is associated with greater renal tissue preservation and favorable postoperative renal function [74]. Radical nephrectomy is related to the increased risk of developing renal disease, which underlies the relationship between RCC and ESKD.

According to EAU guidelines for RCC management, partial nephrectomy should be the mainstay of treatment for T1 tumors and should be preferred, if it is feasible, in patients with T2 tumors and solitary kidney or renal disease [40]. Compared with radical nephrectomy, the advantage of partial nephrectomy concerns preventing CKD in patients with small resectable tumors after the preservation of renal parenchyma [43]. Especially in patients with a solitary kidney that have undergone a partial nephrectomy, a larger remaining renal parenchyma or functional volume comprise independent predictors of renal function postoperatively [75,76]. Of note, a 5% increase in the proportion of remaining kidney is associated with a 17% reduction in the risk of de novo end-stage renal disease [77]. In a study of 1169 patients that underwent either open or laparoscopic partial nephrectomy, an increase in the time of warm ischemia beyond 20 min was related to worse renal function [48]. Interestingly, the lower point of eGFR postoperatively was related to the progression to CKD after adjusting for other risk factors [48]. However, the remaining renal function after prolonged ischemic time (over 30 min) during partial nephrectomy is considered superior to the remaining function after radical surgery [48].

Current evidence regarding the non-invasive and minimally invasive management of RCC in patients with CKD and ESKD is limited but evolving. Available therapeutic modalities include active surveillance, percutaneous thermal ablation (cryoablation, radiofrequency ablation, and microwave ablation), and non-thermal focal techniques such as irreversible electroporation (IRE), and stereotactic body radiotherapy (SBRT). These options are attractive in patients with a compromised renal reserve because they aim to achieve local tumor control while minimizing the loss of functional parenchyma and perioperative morbidity [40]. According to European Guidelines, active surveillance or ablative techniques should be offered in frail patients with comorbidities with small renal masses [40]. Active surveillance is widely endorsed for small renal masses in frail patients or those with a limited life expectancy. In CKD/ESKD populations it offers the advantage of avoiding immediate intervention and preserving renal function, but requires rigorous imaging follow-up and clear criteria for intervention since tumor biology can be heterogeneous and multifocal disease is more common in dialysis populations [78]. Percutaneous thermal ablation has the most substantive clinical experience in CKD/ESKD cohorts. Reported local control rates are high in published studies, and cryoablation, radiofrequency, and microwave techniques have all been used successfully for lesions typically ≤3–4 cm [40,79]. Although local tumor control rates in patients with ESKD undergoing tumor ablation is estimated at 88–100%, only 2–10% of patients with ESKD are reported to undergo these non-invasive treatment options [80]. The effect of ablation treatment in patients with ESKD remain unclear, since the worsening of kidney function may be attributed to both the destruction of renal tissue caused by cryoablation along with the natural course of advanced CKD [80]. In dialysis-dependent patients, measuring the treatment-related decline in renal function is challenging, and the observed trajectory often reflects a combination of procedure-related parenchymal loss and underlying CKD progression [81]. Further well-designed clinical trials are needed to evaluate the role of ablative techniques in maintaining renal function in patients with ESKD.

IRE and other non-thermal approaches provide theoretical advantages for lesions adjacent to critical structures and may spare connective tissue architecture, but clinical experience is more limited and the long-term oncologic outcomes in CKD populations remain unestablished [40,82]. SBRT has emerged as a non-invasive alternative for patients that are poor surgical candidates; early studies suggest promising local control with low acute toxicity, but data specific to CKD/ESKD patients and effects on renal function are still sparse [83]. The following technical and perioperative considerations are particularly important in this population: minimizing nephrotoxic contrast exposure through MRI or ultrasound guidance and CT-US fusion, careful planning to preserve as much viable parenchyma as possible, coordinated periprocedural anticoagulation management in patients on dialysis, and the timing of dialysis sessions around procedures [84]. Pre-treatment biopsy is recommended when feasible to confirm histology and guide management, especially given the higher prevalence of benign and indolent lesions among small renal masses.

In summary, non-invasive and minimally invasive therapies offer promising options for RCC management in patients with CKD/ESKD, balancing oncologic control against the preservation of renal function. Current practice is guided largely by observational data and expert consensus; high-quality prospective trials, standardized outcome measures (including dialysis-free survival and CKD-appropriate renal metrics), and registries are required to define optimal patient selection, technique, and long-term outcomes.

Advanced or metastatic RCC is associated with worse overall survival and requires systematic treatment [80]. During the last decade, cytokine therapy, mainly including interferon and interleukin-2, comprised the first-line option for these patients [85]. Recent well-designed trials have demonstrated that targeted molecular therapies, including VEGF and m-TOR inhibitors, were proved to have equal effectiveness with cytokine-based treatment for advanced RCC, but without the related toxicity [74,86]. In addition, since RCC is considered an immunologically active malignancy, immune checkpoint inhibitors (ICIs) are also considered as a first-line treatment option for these patients [40]. However, the administration of ICIs may exacerbate proteinuria, nephrotic syndrome, and diarrhea in these patients, resulting in worse renal function [42,87]. Targeted therapies and ICIs have demonstrated improved efficacy and tolerability compared with historical cytokine treatments, but each class carries renal-specific toxicities: TKIs and mTOR inhibitors can provoke hypertension and proteinuria, while ICIs can cause immune-related nephritis, nephrotic syndrome, and diarrhea that may accelerate the loss of renal function. Dose modification and careful pharmacokinetic consideration are required in patients with impaired renal function, and the close monitoring of serum creatinine, eGFR, and proteinuria is essential during treatment. Concerning RCC patients with renal transplantation for ESKD, ICC treatment is associated with worse outcomes on graft survival and an increased incidence of graft rejection [40]. ICC treatment should not be used in these patients, since published studies are scarce and these patients are usually excluded from clinical trials. The incorporation of mTOR inhibitors to curb tumor progression in kidney transplant recipients (KTRs) appears promising with respect to overall survival, although definitive outcomes for complex RCC remain undetermined [88]. The use of mTOR inhibitors in KTRs necessitates close surveillance, since these agents have been associated with impaired wound repair and should therefore be avoided before oncologic surgical procedures for RCC or prior to transplantation. A personalized, multidisciplinary approach involving oncology, nephrology, and transplantation specialists is therefore critical to optimize oncologic outcomes while preserving renal function.

## 6. Future Directions

As already mentioned, renal tumors that develop in patients with ESRD or kidney transplant recipients are usually found at a younger age, present with multifocal lesions that are smaller in size, and have less metastatic potential compared with sporadic RCC [43]. The exact pathophysiology by which the uremic state, chronic kidney disease, and proteinuria found in these patients can cause malignant transformation in the kidney still remains unknown. Due to the multifactorial relationship and numerous different pathways, experimental animal models have failed to properly identify the risk factors and molecular pathways. Animal models and human studies should aim to discover the pathophysiological mechanisms.

Screening for traditional tumor markers in patients with chronic kidney disease (CKD) is a controversial but potentially useful approach for cancer monitoring in these patients and could play a significant role in the diagnosis of RCC. Although these markers, such as carcinoembryonic antigen (CEA), cancer antigen 125 (CA-125), and prostate-specific antigen (PSA), can be elevated in CKD patients due to impaired renal clearance, their trends over time may still provide valuable information [89]. An unexplained, persistent rise in these markers could signal the presence of a developing malignancy, prompting further investigation. Given the heightened cancer risk in this population, exploring novel and more specific biomarkers for early cancer detection is crucial [90,91]. These new biomarkers should ideally not be affected by the uremic environment, providing a more reliable signal. Developing a panel of CKD-specific biomarkers could enable earlier diagnosis, leading to improved outcomes for these vulnerable patients.

Future minimally invasive therapies for RCC in CKD and ESRD patients will prioritize maximal tumor control while preserving residual renal function. Advances in image-guided percutaneous ablation (cryoablation, radiofrequency, and microwave) and nonthermal modalities like irreversible electroporation, combined with real-time CT/MRI–ultrasound fusion and robotic assistance, are expected to improve precision and safety. Artificial intelligence-driven treatment planning and intraoperative monitoring will enable personalized, adaptive therapies with shorter recovery times and lower complication rates. Well-designed prospective trials with long-term renal functional and oncologic endpoints are essential to define optimal protocols and confirm that these approaches reduce the progression to ESRD without compromising cancer control.

## 7. Limitations

Our review is subject to several limitations. As already mentioned, renal tumors that develop in patients with ESRD or kidney transplant recipients are usually found at a younger age, present with multifocal lesions that are smaller in size, and have less metastatic potential compared with sporadic RCC [43]. The exact pathophysiology by which the uremic state, chronic kidney disease, and proteinuria found in these patients can cause malignant transformation in the kidney still remains unknown. Due to the multifactorial relationship and numerous different pathways, experimental animal models have failed to properly identify the risk factors and molecular pathways. In addition, our study comprises a narrative review of the literature. Although thorough research of the literature was made with specific inclusion and exclusion criteria, it does not follow the structure of a systematic review.

## 8. Conclusions

The relationship between RCC and renal dysfunction is bidirectional and multifactorial. Apart from the progression of a tumor itself and the advancement of renal disease in these patients, several risk factors, including hypertension, diabetes, obesity, and smoking may predispose and contribute to the worsening of both CKD and RCC in these frail patients. This evolving condition along with the associated pathophysiological mechanisms should be brought to the attention of urologists and should determine the treatment options for these patients by selecting mainly nephron-sparing approaches. Further well-designed studies are needed to elucidate the most suitable treatment option for these patients and clarify the role of minimally invasive therapies, such as ablative techniques.

## Figures and Tables

**Figure 1 biomedicines-13-02638-f001:**
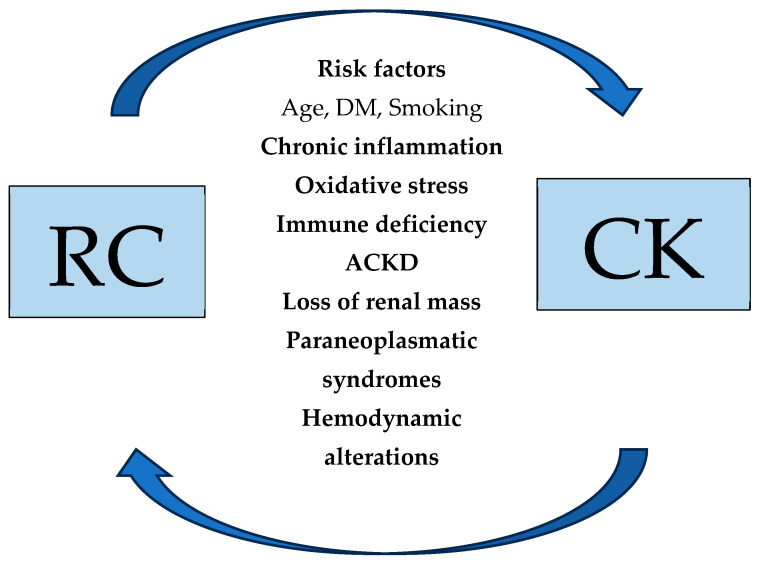
Bidirectional relationship between RCC and CKD development.

**Figure 2 biomedicines-13-02638-f002:**
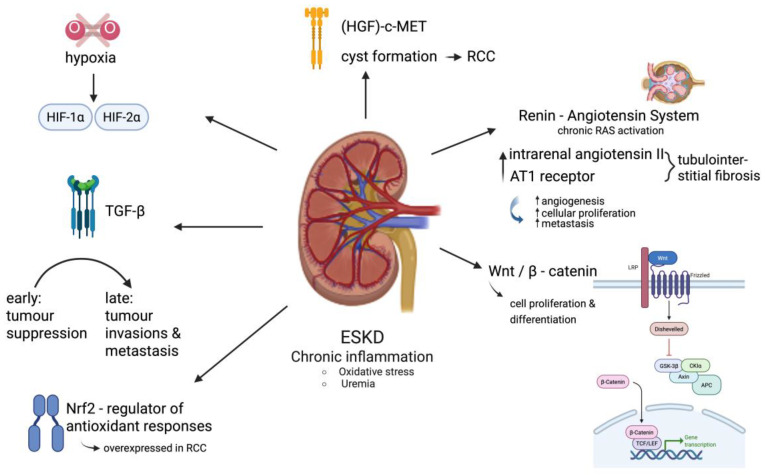
Molecular pathways of the relationship between RCC development and CKD.

**Table 1 biomedicines-13-02638-t001:** Risk factors for the development of RCC in CKD patients and CKD in RCC patients. RCC: Renal Cell Cancer, CKD: Chronic Kidney Disease.

Risk Factors for the Development	
**RCC in CKD patients**	**CKD in RCC patients**
Acquired cystic disease in ESKD patients	**Patient characteristics**
Renal injury	Age, Sex, Ethnicity
Vascular injury	**Lifestyle** Smoking
	**Other Comorbidities**
	DM, Hypertension, Genetic factors
	**Pre-surgical renal factors**
	Proteinuria, Renal Disease, Interstitial fibrosis, Tubular Atrophy, Arteriolar Sclerosis
	**Surgical factors**
	Surgical Technique, Tumor Volume

## Data Availability

Data sharing is not applicable.

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
