# Peer review of "Association Between Renal Cell Cancer and Chronic Kidney Disease: An Update on a Never-Healing Wound"

_biomedicines, 2025, doi:10.3390/biomedicines13112638_

Round 1

Reviewer 1 Report

Comments and Suggestions for Authors

Ilias Giannakodimos et al. conducted a review that is to update the knowledge of the Association between Renal Cell Cancer and Chronic Kidney Disease; An update of a never-healing wound. This topic is clinically important and worth investigating; however, there are caveats in the present study as indicated below.
1)    The English is of good quality with only minor issues that do not impede comprehension. With light editing, the manuscript would achieve excellent linguistic standards. I recommend careful proofreading to catch remaining typos and ensure consistency in terminology.
2)    More follow-up information, including the median and range of follow-up period and the number of events, should be described in the manuscript.
3)    Clinicopathological information should also be described
4)    More information about systematic treatment
5)    I suggest slightly expanding the limitations section to highlight the absence of experimental validation and the need for future confirmation. 
Overall, this is a solid manuscript that can be accepted after addressing the above minor revisions. I look forward to receiving your revised version.

Author Response

Ilias Giannakodimos et al. conducted a review that is to update the knowledge of the Association between Renal Cell Cancer and Chronic Kidney Disease; An update of a never-healing wound. This topic is clinically important and worth investigating; however, there are caveats in the present study as indicated below.

1)    The English is of good quality with only minor issues that do not impede comprehension. With light editing, the manuscript would achieve excellent linguistic standards. I recommend careful proofreading to catch remaining typos and ensure consistency in terminology.

Thank you for your valuable remarks. Careful proofreading was done and several expressions were changed, while keeping the same meaning, as you proposed.

2)    More follow-up information, including the median and range of follow-up period and the number of events, should be described in the manuscript.

Thank you for your valuable comment. Follow-up information, regarding follow-up period and number of events were added in the text, especially in Epidemiology and Risk factors sections.

3)    Clinicopathological information should also be described

Thank you for your interesting remark. Although several clinicopathological details are already described in the manuscript regarding the development of different types of renal cancers, additional clinicopathological information were added in the text, mainly in Risk factor section.

4)    More information about systematic treatment

Thank you for your interesting comment. The following text was added concerning systematic treatment: “Targeted therapies and ICIs have demonstrated improved efficacy and tolerability compared with historical cytokine treatments, but each class carries renal‑specific toxicities: TKIs and mTOR inhibitors can provoke hypertension and proteinuria, while ICIs can cause immune‑related nephritis, nephrotic syndrome and diarrhea that may accelerate loss of renal function. Dose modification and careful pharmacokinetic consideration are required in patients with impaired renal function, and close monitoring of serum creatinine, eGFR and proteinuria is essential during treatment.” And “The incorporation of mTOR inhibitors to curb tumor progression in kidney transplant recipients (KTRs) appears promising with respect to overall survival, although definitive outcomes for complex RCC remain undetermined (90). Use of mTOR inhibitors in KTRs necessitates close surveillance, since these agents have been associated with impaired wound repair and should therefore be avoided before oncologic surgical procedures for RCC or prior to transplantation. A personalized, multidisciplinary approach involving oncology, nephrology and transplantation specialists is therefore critical to optimize oncologic outcomes while preserving renal function.”

5)    I suggest slightly expanding the limitations section to highlight the absence of experimental validation and the need for future confirmation.

Thank you for your interesting comment. The limitation section was expanded as you proposed and a Limitation section was added, highlighting experimental validation: “Our review is subjected to several limitations. As already mentioned, renal tumors that develop in patients with ESRD or kidney transplant recipients are usually found in younger age, present with multifocal lesions are smaller in size, and have less metastatic potential compared to sporadic RCC (43). The exact pathophysiology by which the uremic state, chronic kidney disease and proteinuria found in these patients, can cause malignant transformation in the kidney still remains unknown. Due to multifactorial relationship and numerous different pathways, experimental animal models have failed to properly identify risk factors and molecular pathways. In addition, our study comprises a narrative review of the literature. Although, thorough research of the literature was made with specific inclusion and exclusion criteria, it does not follow the structure of a systematic review.”

Additionally, the need for future confirmation is highlighted in the Future direction section. The following paragraphs were added: “Screening for traditional tumor markers in patients with chronic kidney disease (CKD) is a controversial but potentially useful approach for cancer monitoring in these patients and could play a significant role for the diagnosis of RCC. Although these markers, such as carcinoembryonic antigen (CEA), cancer antigen 125 (CA-125), and prostate-specific antigen (PSA), can be elevated in CKD patients due to impaired renal clearance, their trends over time may still provide valuable information (91). An unexplained, persistent rise in these markers could signal the presence of a developing malignancy, prompting further investigation. Given the heightened cancer risk in this population, exploring novel and more specific biomarkers for early cancer detection is crucial. These new biomarkers should ideally not be affected by the uremic environment, providing a more reliable signal.  Developing a panel of CKD-specific biomarkers could enable earlier diagnosis, leading to improved outcomes for these vulnerable patients. Future minimally invasive therapies for RCC in CKD and ESRD patients will prioritize maximal tumor control while preserving residual renal function. Advances in image-guided percutaneous ablation (cryoablation, radiofrequency, microwave) and nonthermal modalities like irreversible electroporation, combined with real-time CT/MRI–ultrasound fusion and robotic assistance, are expected to improve precision and safety. Artificial intelligence–driven treatment planning and intraoperative monitoring will enable personalized, adaptive therapies with shorter recovery times and lower complication rates. Well-designed prospective trials with long-term renal functional and oncologic endpoints are essential to define optimal protocols and confirm that these approaches reduce progression to ESRD without compromising cancer control.”

Overall, this is a solid manuscript that can be accepted after addressing the above minor revisions. I look forward to receiving your revised version.

Reviewer 2 Report

Comments and Suggestions for Authors

This manuscript aims to explore the bidirectional relationships between CKD and renal cancer, as well as the potential risk factors underlying them and their pathophysiological mechanisms, as a significant step toward a new treatment option suitable for these patients.

The manuscript is generally well written and covers this specific topic. However, to increase the quality of the manuscript, I have a few suggestions for the authors:

- Albuminuria is associated with a higher risk of death from renal cancers. Please elaborate on this feature in section 2.2 of the manuscript.

- Activation of the Wnt/β-catenin signalling pathway is also a significant step in the CKD-related inflammatory/ROS system and cancer development. In addition, the transcription factor Nrf2 played a significant role in antioxidant responses in patients with CKD. Although limited data are illustrated in section 2.3.3, the authors should add additional data regarding their potential mechanisms for increasing cancer risk in patients with CKD.

- Nitrogen-containing substances and carcinogenic compounds accumulate in the blood of patients with ESKD, which increases the risk of cancer in these patients. Excessive parathyroid hormone and vitamin D deficiency may also be pro-carcinogenic factors in ESKD patients. Please add additional data on their potential role in renal carcinogenesis in these cases.

- Although controversial, screening for traditional tumour markers may be used as cancer monitoring in patients with CKD. Specific biomarkers for early cancer screening in these patients also require further investigation. However, this aspect should be briefly described by the authors.

- Kindly add the limitations of this study at the end of Section 4.

- Please edit the reference list according to the recommendations of the Biomedicines journal.

Author Response

Reviewer 2

This manuscript aims to explore the bidirectional relationships between CKD and renal cancer, as well as the potential risk factors underlying them and their pathophysiological mechanisms, as a significant step toward a new treatment option suitable for these patients.

The manuscript is generally well written and covers this specific topic. However, to increase the quality of the manuscript, I have a few suggestions for the authors:

Thank you for your inspiring comments

- Albuminuria is associated with a higher risk of death from renal cancers. Please elaborate on this feature in section 2.2 of the manuscript.

Thank you for your informative addition. The following paragraph was added: “Interestingly, although albuminuria comprised a known sign of renal damage and progression to CKD, it may also reflect microvascular damage or affected glomerular permeability, caused by tumor products (25). Increased levels of urine albumin are related with advanced tumor stages and higher tumor burden that are significant factors for cancer related death (26). As a result, albuminuria may reflect a worse systematic or tumor burden and is related with increased risk of cancer death in RCC patients, associated with worse survival and more aggressive disease (26).”

- Activation of the Wnt/β-catenin signalling pathway is also a significant step in the CKD-related inflammatory/ROS system and cancer development. In addition, the transcription factor Nrf2 played a significant role in antioxidant responses in patients with CKD. Although limited data are illustrated in section 2.3.3, the authors should add additional data regarding their potential mechanisms for increasing cancer risk in patients with CKD.

Thank you for your informative comment. The following paragraphs were added, as you proposed: “Wnt/β-catenin activation is a well-established oncogenic driver. More specifically, the accumulation of β-catenin in the nucleus leads to the transcription of target genes, such as c-Myc and cyclin D1, which promote uncontrolled cell growth and inhibit apoptosis. The chronic inflammatory state in CKD can activate this pathway, creating a pro-carcinogenic environment.” And “In the presence of oxidative stress, Nrf2 is released, translocates to the nucleus, and binds to antioxidant response elements (AREs), promoting the transcription of antioxidant and detoxification genes. While Nrf2 activation can initially suppress cancer development by protecting against DNA damage, a sustained and chronic activation can make cancer cells resistant to chemotherapy and radiotherapy.  The interplay between the Wnt/β-catenin and Nrf2 pathways is a key area of research. Of note, several studies suggest that Wnt/β-catenin can modulate Nrf2 activity, and vice versa, which can contribute to the complex relationship between CKD and cancer.”

- Nitrogen-containing substances and carcinogenic compounds accumulate in the blood of patients with ESKD, which increases the risk of cancer in these patients. Excessive parathyroid hormone and vitamin D deficiency may also be pro-carcinogenic factors in ESKD patients. Please add additional data on their potential role in renal carcinogenesis in these cases.

Thank you for your valuable comment. According to your interesting contribution, the following sentences were added “Another mechanism suggests that hyperparathyroidism in ESKD patients contributes to renal carcinogenesis by promoting unregulated cell proliferation in the kidneys. High levels of Parathyroid Hormone (PTH) can act as a growth factor, leading to the rapid division of cells and increasing the likelihood of genetic mutations (58). This hormone also worsens the body's already impaired DNA repair mechanisms in uremic patients, allowing damage to accumulate (58). Furthermore, PTH-induced hypercalcemia can create an inflammatory environment in the kidneys, which is a known risk factor for cancer development. Together, these processes create a pro-carcinogenic environment, making the kidneys more susceptible to tumor formation (58).”, “A profound deficiency of active vitamin D, or calcitriol, is common in ESKD patients and removes a critical anti-cancer defense mechanism. Calcitriol normally has powerful antiproliferative effects, meaning it helps prevent the uncontrolled growth of cells (72).  Without it, kidney cells are more prone to becoming cancerous. The deficiency also impairs cellular differentiation, leading to immature, cancer-like cells that can more easily form tumors (72). This loss of normal gene regulation creates a highly favorable environment for the initiation and progression of cancer (72).” And “Another key mechanism for carcinogenesis is the formation of N-nitroso compounds (NOCs), which are potent carcinogens, since the kidneys of ESKD patients fail to excrete nitrogenous waste, leading to the accumulation of uremic toxins (67). The high concentration of nitrogenous compounds in the blood promotes their conversion into these cancer-causing agents (67). This buildup also contributes to chronic inflammation and oxidative stress, which directly damage cellular DNA and impair the body's ability to repair itself. This continuous assault on kidney cells from multiple fronts significantly increases the risk of renal cancer.”

- Although controversial, screening for traditional tumour markers may be used as cancer monitoring in patients with CKD. Specific biomarkers for early cancer screening in these patients also require further investigation. However, this aspect should be briefly described by the authors.

Thank you for your valuable comment. In order to analyze the role of biomarkers in patients with CKD, the following sentences were added in the Future Directions section; “Screening for traditional tumor markers in patients with chronic kidney disease (CKD) is a controversial but potentially useful approach for cancer monitoring in these patients and could play a significant role for the diagnosis of RCC. Although these markers, such as carcinoembryonic antigen (CEA), cancer antigen 125 (CA-125), and prostate-specific antigen (PSA), can be elevated in CKD patients due to impaired renal clearance, their trends over time may still provide valuable information (89). An unexplained, persistent rise in these markers could signal the presence of a developing malignancy, prompting further investigation. Given the heightened cancer risk in this population, exploring novel and more specific biomarkers for early cancer detection is crucial. These new biomarkers should ideally not be affected by the uremic environment, providing a more reliable signal.  Developing a panel of CKD-specific biomarkers could enable earlier diagnosis, leading to improved outcomes for these vulnerable patients.

- Kindly add the limitations of this study at the end of Section 4.

Thank you for your constructive comment. The limitations were moved at the end of section 4, as you proposed and the following paragraph was added: “Our review is subjected to several limitations. As already mentioned, renal tumors that develop in patients with ESRD or kidney transplant recipients are usually found in younger age, present with multifocal lesions are smaller in size, and have less metastatic potential compared to sporadic RCC (43). The exact pathophysiology by which the uremic state, chronic kidney disease and proteinuria found in these patients, can cause malignant transformation in the kidney still remains unknown. Due to multifactorial relationship and numerous different pathways, experimental animal models have failed to properly identify risk factors and molecular pathways. In addition, our study comprises a narrative review of the literature. Although, thorough research of the literature was made with specific inclusion and exclusion criteria, it does not follow the structure of a systematic review.”

- Please edit the reference list according to the recommendations of the Biomedicines journal.

The reference list was edited as you proposed.

Reviewer 3 Report

Comments and Suggestions for Authors
  1. The purpose of the study is not clearly stated, authors should clearly define the aim and research questions in the introduction.
  2. Literature review is brief and misses some recent studies, please add 2 to 3 recent studies (last 5 years) to strengthen the rationale.
  3. Provide inclusion and exclusion criteria in detail.
  4. Discussion is descriptive and does not critically compare findings with past work. Authors should add a short paragraph on practical or clinical implications.
  5. Avoid overgeneralization and link conclusion strictly to your data. The conclusion makes broad claims not fully supported by the data.

Author Response

This manuscript aims to explore the bidirectional relationships between CKD and renal cancer, as well as the potential risk factors underlying them and their pathophysiological mechanisms, as a significant step toward a new treatment option suitable for these patients.

The manuscript is generally well written and covers this specific topic. However, to increase the quality of the manuscript, I have a few suggestions for the authors:

Thank you for your inspiring comments

- Albuminuria is associated with a higher risk of death from renal cancers. Please elaborate on this feature in section 2.2 of the manuscript.

Thank you for your informative addition. The following paragraph was added: “Interestingly, although albuminuria comprised a known sign of renal damage and progression to CKD, it may also reflect microvascular damage or affected glomerular permeability, caused by tumor products (25). Increased levels of urine albumin are related with advanced tumor stages and higher tumor burden that are significant factors for cancer related death (26). As a result, albuminuria may reflect a worse systematic or tumor burden and is related with increased risk of cancer death in RCC patients, associated with worse survival and more aggressive disease (26).”

- Activation of the Wnt/β-catenin signalling pathway is also a significant step in the CKD-related inflammatory/ROS system and cancer development. In addition, the transcription factor Nrf2 played a significant role in antioxidant responses in patients with CKD. Although limited data are illustrated in section 2.3.3, the authors should add additional data regarding their potential mechanisms for increasing cancer risk in patients with CKD.

Thank you for your informative comment. The following paragraphs were added, as you proposed: “Wnt/β-catenin activation is a well-established oncogenic driver. More specifically, the accumulation of β-catenin in the nucleus leads to the transcription of target genes, such as c-Myc and cyclin D1, which promote uncontrolled cell growth and inhibit apoptosis. The chronic inflammatory state in CKD can activate this pathway, creating a pro-carcinogenic environment.” And “In the presence of oxidative stress, Nrf2 is released, translocates to the nucleus, and binds to antioxidant response elements (AREs), promoting the transcription of antioxidant and detoxification genes. While Nrf2 activation can initially suppress cancer development by protecting against DNA damage, a sustained and chronic activation can make cancer cells resistant to chemotherapy and radiotherapy.  The interplay between the Wnt/β-catenin and Nrf2 pathways is a key area of research. Of note, several studies suggest that Wnt/β-catenin can modulate Nrf2 activity, and vice versa, which can contribute to the complex relationship between CKD and cancer.”

- Nitrogen-containing substances and carcinogenic compounds accumulate in the blood of patients with ESKD, which increases the risk of cancer in these patients. Excessive parathyroid hormone and vitamin D deficiency may also be pro-carcinogenic factors in ESKD patients. Please add additional data on their potential role in renal carcinogenesis in these cases.

Thank you for your valuable comment. According to your interesting contribution, the following sentences were added “Another mechanism suggests that hyperparathyroidism in ESKD patients contributes to renal carcinogenesis by promoting unregulated cell proliferation in the kidneys. High levels of Parathyroid Hormone (PTH) can act as a growth factor, leading to the rapid division of cells and increasing the likelihood of genetic mutations (58). This hormone also worsens the body's already impaired DNA repair mechanisms in uremic patients, allowing damage to accumulate (58). Furthermore, PTH-induced hypercalcemia can create an inflammatory environment in the kidneys, which is a known risk factor for cancer development. Together, these processes create a pro-carcinogenic environment, making the kidneys more susceptible to tumor formation (58).”, “A profound deficiency of active vitamin D, or calcitriol, is common in ESKD patients and removes a critical anti-cancer defense mechanism. Calcitriol normally has powerful antiproliferative effects, meaning it helps prevent the uncontrolled growth of cells (72).  Without it, kidney cells are more prone to becoming cancerous. The deficiency also impairs cellular differentiation, leading to immature, cancer-like cells that can more easily form tumors (72). This loss of normal gene regulation creates a highly favorable environment for the initiation and progression of cancer (72).” And “Another key mechanism for carcinogenesis is the formation of N-nitroso compounds (NOCs), which are potent carcinogens, since the kidneys of ESKD patients fail to excrete nitrogenous waste, leading to the accumulation of uremic toxins (67). The high concentration of nitrogenous compounds in the blood promotes their conversion into these cancer-causing agents (67). This buildup also contributes to chronic inflammation and oxidative stress, which directly damage cellular DNA and impair the body's ability to repair itself. This continuous assault on kidney cells from multiple fronts significantly increases the risk of renal cancer.”

- Although controversial, screening for traditional tumour markers may be used as cancer monitoring in patients with CKD. Specific biomarkers for early cancer screening in these patients also require further investigation. However, this aspect should be briefly described by the authors.

Thank you for your valuable comment. In order to analyze the role of biomarkers in patients with CKD, the following sentences were added in the Future Directions section; “Screening for traditional tumor markers in patients with chronic kidney disease (CKD) is a controversial but potentially useful approach for cancer monitoring in these patients and could play a significant role for the diagnosis of RCC. Although these markers, such as carcinoembryonic antigen (CEA), cancer antigen 125 (CA-125), and prostate-specific antigen (PSA), can be elevated in CKD patients due to impaired renal clearance, their trends over time may still provide valuable information (89). An unexplained, persistent rise in these markers could signal the presence of a developing malignancy, prompting further investigation. Given the heightened cancer risk in this population, exploring novel and more specific biomarkers for early cancer detection is crucial. These new biomarkers should ideally not be affected by the uremic environment, providing a more reliable signal.  Developing a panel of CKD-specific biomarkers could enable earlier diagnosis, leading to improved outcomes for these vulnerable patients.

- Kindly add the limitations of this study at the end of Section 4.

Thank you for your constructive comment. The limitations were moved at the end of section 4, as you proposed and the following paragraph was added: “Our review is subjected to several limitations. As already mentioned, renal tumors that develop in patients with ESRD or kidney transplant recipients are usually found in younger age, present with multifocal lesions are smaller in size, and have less metastatic potential compared to sporadic RCC (43). The exact pathophysiology by which the uremic state, chronic kidney disease and proteinuria found in these patients, can cause malignant transformation in the kidney still remains unknown. Due to multifactorial relationship and numerous different pathways, experimental animal models have failed to properly identify risk factors and molecular pathways. In addition, our study comprises a narrative review of the literature. Although, thorough research of the literature was made with specific inclusion and exclusion criteria, it does not follow the structure of a systematic review.”

- Please edit the reference list according to the recommendations of the Biomedicines journal.

The reference list was edited as you proposed.

Reviewer 3

The purpose of the study is not clearly stated, authors should clearly define the aim and research questions in the introduction.

Thank you for your constructive remark. The following sentence was added in the Introduction section: “Also, the main purpose of this review was to describe the proper management of RCC cases in ESKD patient and provide future directions for this issue.”

Literature review is brief and misses some recent studies, please add 2 to 3 recent studies (last 5 years) to strengthen the rationale.

Thank you for your valuable remark. More than 3 recently published studies were added as you proposed.

Provide inclusion and exclusion criteria in detail.

Thank you for your interesting comment. Regarding inclusion and exclusion criteria, the Literature search section was added and the following paragraphs were added: “We reviewed the current literature from the inception of this current review until August 2025. For our scope, we used “PubMed” database and we included only studies written in English. The search was based on the following search terms: “Renal cell cancer”, “renal disease”, “End-stage renal disease”, “chronic kidney disease” and “transplant recipients” and we retrieved the results of our search for the epidemiology, pathophysiologic mechanisms, risk factor and treatment sections. Also, the references of the research articles were scrutinized for relevant studies.

Only studies that concerned human adults (≥18 years) with CKD, ESKD and/or diagnosed RCC were included. Retrospective or prospective studies, epidemiological research, systematic reviews and/or meta-analyses that concerned CKD or ESKD patients (dialysis or transplant) or patients with RCC were included in our study . RCC cases were defined by histological diagnosis or validated cancer registry coding. Studies with interventions related to RCC management (partial/radical nephrectomy, ablation, systemic therapies including immunotherapy/targeted agents) and their renal effects were also included. Finally, studies that referred to Pediatric populations (<18 years), do not define CKD or RCC using standard clinical or histologic criteria, case reports and non-English literature were excluded.”

Discussion is descriptive and does not critically compare findings with past work. Authors should add a short paragraph on practical or clinical implications.

Thank you for your interesting addition. Practical implications of our study are discusses in Future directions section, since in concerns the unsolved problems of our research and what we expect in the future.

Avoid overgeneralization and link conclusion strictly to your data. The conclusion makes broad claims not fully supported by the data.

Thank you for your constructive comment. Further data of included studies were added in order to avoid overegeneralization and make conclusions more consistent.

Reviewer 4 Report

Comments and Suggestions for Authors

The current study is on a topic of general interest to the readers of the journal, and a valuable resource for understanding the complex interplay between RCC and CKD. However, the review article needs more concise writing, enhanced visuals, and updated references. So I could not recommend this paper for publication without major revisions.

Major comments:

  1. The article lacks novel insights; it primarily updates existing knowledge but does not introduce groundbreaking findings or perspectives.
  2. The article suffers from repetition, so authors should reduce redundancy and focus on presenting concise, impactful information:
  • Some concepts, such as the bidirectional relationship between RCC and CKD, are repeated multiple times, which could be streamlined for better readability.
  • In page 2; this paragraph “Finally, the association of RCC development in transplant recipient patients is also considered unique. Although kidney transplant recipients (KTRs) carry an elevated cancer risk in general, due to immunosuppression and oncogenic viruses, RCC constitutes the most common post-transplant malignancy involving the urinary system.” Also, “Risk factors for RCC in KTRs include the same factors as in dialysis patients; long pre-transplant dialysis duration, ACKD and immunosuppressive treatment (3). Due to elevated risk of RCC, lifelong surveillance of native kidneys is recommended in transplant recipients with risk factors, especially in patients with a prolonged dialysis history before transplant (11).” These two paragraphs should be moved from the epidemiological factors to risk factors
  • In page 3, 2nd line, the authors repeated exactly the same sentence “As already mentioned, the risk of RCC development among patients with CKD varies from no risk up to 100-fold;” which has been mentioned under title “Epidemiology of RCC in Patients with Renal Disease”. Remove repeated information
  • The authors keep repeating in page 3 “as already mentioned”, if the information has been mentioned, why is the repetition
  1. The article has Limited Visuals: While Figure 1 is mentioned, additional diagrams or flowcharts summarizing molecular pathways or treatment options could enhance understanding:
  • The authors mentioned in the introduction that they demonstrated “the bidirectional relationship between RCC development and occurrence of CKD is demonstrated in Figure 1”. After looking for Figure 1, it appeared at the end of the review article as a co-shared figure with “future directions”. This is very annoying and confusing. It has to be removed from “future directions” and be placed in its allocated place in the introduction
  • Table 1 neither adds any added value to the review article, nor is it well-structured
  1. The discussion on non-invasive treatments for RCC in CKD patients is brief and lacks robust evidence or detailed analysis.
  2. Future Directions need to be expanded and should provide more specific recommendations for research, particularly regarding minimally invasive therapies and molecular mechanisms. ​
  3. Some references are outdated; updating the references will strengthen the review and will reflect the latest advancements in the field

Author Response

Reviewer 4

The current study is on a topic of general interest to the readers of the journal, and a valuable resource for understanding the complex interplay between RCC and CKD. However, the review article needs more concise writing, enhanced visuals, and updated references. So I could not recommend this paper for publication without major revisions.

Thank you for your interesting comments. Major chances were made in the initial text as you proposed.

The article lacks novel insights; it primarily updates existing knowledge but does not introduce groundbreaking findings or perspectives.

Thank you for your important notice. The section “Future directions” was expanded in order to provide novel insights.

The article suffers from repetition, so authors should reduce redundancy and focus on presenting concise, impactful information:

Some concepts, such as the bidirectional relationship between RCC and CKD, are repeated multiple times, which could be streamlined for better readability.

Thank you for your interesting comment. The phrase “bidirectional relationship” was erased from the main text, when possible, to avoid repetition, as you proposed.

In page 2; this paragraph “Finally, the association of RCC development in transplant recipient patients is also considered unique. Although kidney transplant recipients (KTRs) carry an elevated cancer risk in general, due to immunosuppression and oncogenic viruses, RCC constitutes the most common post-transplant malignancy involving the urinary system.” Also, “Risk factors for RCC in KTRs include the same factors as in dialysis patients; long pre-transplant dialysis duration, ACKD and immunosuppressive treatment (3). Due to elevated risk of RCC, lifelong surveillance of native kidneys is recommended in transplant recipients with risk factors, especially in patients with a prolonged dialysis history before transplant (11).” These two paragraphs should be moved from the epidemiological factors to risk factors

Thank you for your constructive comments. The following changes in paragraph formation were made as you proposed.

In page 3, 2nd line, the authors repeated exactly the same sentence “As already mentioned, the risk of RCC development among patients with CKD varies from no risk up to 100-fold;” which has been mentioned under title “Epidemiology of RCC in Patients with Renal Disease”. Remove repeated information

Thank you for your valuable remark. The repeated information was erased as you proposed.

The authors keep repeating in page 3 “as already mentioned”, if the information has been mentioned, why is the repetition

Thank you for your interesting remark. The phrase “as already mentioned” was used to connect paragraphs together and was not used due to repetition of meaning, since new information were added. As a result, the phrase was erased as you mentioned.

The article has Limited Visuals: While Figure 1 is mentioned, additional diagrams or flowcharts summarizing molecular pathways or treatment options could enhance understanding:

The authors mentioned in the introduction that they demonstrated “the bidirectional relationship between RCC development and occurrence of CKD is demonstrated in Figure 1”. After looking for Figure 1, it appeared at the end of the review article as a co-shared figure with “future directions”. This is very annoying and confusing. It has to be removed from “future directions” and be placed in its allocated place in the introduction

Thank you for your constructive comment. Figure 1 was erased for the Future Direction section as you proposed. In addition, Figure 2, summarizing molecular pathways was added, in order to enhance Visuals of our manuscript. In addition the phrase “Molecular pathways in ESKD associated RCC are presented in Figure 2” was also added.

Table 1 neither adds any added value to the review article, nor is it well-structured

Thank you for your valuable comment. However, Table 1 summarizes risk factors for the development of RCC in CKD patients and CKD in RCC patients and make these factors more comprehensive in the audience.

The discussion on non-invasive treatments for RCC in CKD patients is brief and lacks robust evidence or detailed analysis.

Thank you for your interesting comment. The following paragraphs were added in this section: “Current evidence regarding non‑invasive and minimally invasive management of RCC in patients with CKD and ESKD is limited but evolving. Available therapeutic modalities include active surveillance, percutaneous thermal ablation (cryoablation, radiofrequency ablation, microwave ablation), non‑thermal focal techniques such as irreversible electroporation (IRE), and stereotactic body radiotherapy (SBRT). These options are attractive in patients with compromised renal reserve because they aim to achieve local tumor control while minimizing loss of functional parenchyma and perioperative morbidity (40).” “Active surveillance is widely endorsed for small renal masses in frail patients or those with limited life expectancy. In CKD/ESKD populations it offers the advantage of avoiding immediate intervention and preserving renal function, but requires rigorous imaging follow‑up and clear criteria for intervention since tumor biology can be heterogeneous and multifocal disease is more common in dialysis populations (78). These surgical options are considered a possible solution in ESKD disease patients since they can offer complete tumor removal and maintenance of kidney function. Percutaneous thermal ablation has the most substantive clinical experience in CKD/ESKD cohorts. Reported local control rates are high in published series, and cryoablation, radiofrequency and microwave techniques have all been used successfully for lesions typically ≤3–4 cm (79)(40)”, “In dialysis‑dependent patients, measuring treatment‑related decline in renal function is challenging, and the observed trajectory often reflects a combination of procedure‑related parenchymal loss and underlying CKD progression (81).” And “IRE and other non‑thermal approaches provide theoretical advantages for lesions adjacent to critical structures and may spare connective tissue architecture, but clinical experience is more limited and long‑term oncologic outcomes in CKD populations remain unestablished (82)(40). SBRT has emerged as a non‑invasive alternative for patients that are poor surgical candidates; early series suggest promising local control with low acute toxicity, but data specific to CKD/ESKD patients and effects on renal function are still sparse (83). Technical and perioperative considerations are particularly important in this population: minimizing nephrotoxic contrast exposure through MRI or ultrasound guidance and CT‑US fusion, careful planning to preserve as much viable parenchyma as possible, coordinated periprocedural anticoagulation management in patients on dialysis, and timing of dialysis sessions around procedures (84). Pre‑treatment biopsy is recommended when feasible to confirm histology and guide management, especially given the higher prevalence of benign and indolent lesions among small renal masses.

In summary, non‑invasive and minimally invasive therapies offer promising options for RCC management in patients with CKD/ESKD, balancing oncologic control against preservation of renal function. Current practice is guided largely by observational data and expert consensus; high‑quality prospective trials, standardized outcome measures (including dialysis‑free survival and CKD‑appropriate renal metrics), and registries are required to define optimal patient selection, technique, and long‑term outcomes.”

Future Directions need to be expanded and should provide more specific recommendations for research, particularly regarding minimally invasive therapies and molecular mechanisms. ​

Thank you for your valuable comment. The following text was added in the Future Directions section to provide future molecular mechanisms: “Screening for traditional tumor markers in patients with chronic kidney disease (CKD) is a controversial but potentially useful approach for cancer monitoring in these patients and could play an significant role for the diagnosis of RCC. Although these markers, such as carcinoembryonic antigen (CEA), cancer antigen 125 (CA-125), and prostate-specific antigen (PSA), can be elevated in CKD patients due to impaired renal clearance, their trends over time may still provide valuable information (26907957). An unexplained, persistent rise in these markers could signal the presence of a developing malignancy, prompting further investigation. Given the heightened cancer risk in this population, exploring novel and more specific biomarkers for early cancer detection is crucial. These new biomarkers should ideally not be affected by the uremic environment, providing a more reliable signal.  Developing a panel of CKD-specific biomarkers could enable earlier diagnosis, leading to improved outcomes for these vulnerable patients.”

In addition, the following text was added for future perspectives in minimally invasive treatment:

“Future minimally invasive therapies for RCC in CKD and ESRD patients will prioritize maximal tumor control while preserving residual renal function. Advances in image-guided percutaneous ablation (cryoablation, radiofrequency, microwave) and nonthermal modalities like irreversible electroporation, combined with real-time CT/MRI–ultrasound fusion and robotic assistance, are expected to improve precision and safety. Artificial intelligence–driven treatment planning and intraoperative monitoring will enable personalized, adaptive therapies with shorter recovery times and lower complication rates. Well-designed prospective trials with long-term renal functional and oncologic endpoints are essential to define optimal protocols and confirm that these approaches reduce progression to ESRD without compromising cancer control.”

Some references are outdated; updating the references will strengthen the review and will reflect the latest advancements in the field

Thank you for your valuable remark. References were updated as you proposed.

Round 2

Reviewer 2 Report

Comments and Suggestions for Authors

The authors provided a revised version of the manuscript addressing most of the comments.

However, I have only two suggestions for the authors:

  • As the authors mentioned, this paragraph was not included in the main text of the manuscript.

[The following paragraph was added: “Interestingly, although albuminuria comprised a known sign of renal damage and progression to CKD, it may also reflect microvascular damage or affected glomerular permeability, caused by tumor products (25). Increased levels of urine albumin are related with advanced tumor stages and higher tumor burden that are significant factors for cancer related death (26). As a result, albuminuria may reflect a worse systematic or tumor burden and is related with increased risk of cancer death in RCC patients, associated with worse survival and more aggressive disease (26).”]

  • Kindly revise the reference list according to the recommendations of the Biomedicines Journal (e.g., the journal name should be in italics). Please follow the link below: https://www.mdpi.com/journal/biomedicines/instructions

Author Response

The authors provided a revised version of the manuscript addressing most of the comments.

However, I have only two suggestions for the authors:

  • As the authors mentioned, this paragraph was not included in the main text of the manuscript.

[The following paragraph was added: “Interestingly, although albuminuria comprised a known sign of renal damage and progression to CKD, it may also reflect microvascular damage or affected glomerular permeability, caused by tumor products (25). Increased levels of urine albumin are related with advanced tumor stages and higher tumor burden that are significant factors for cancer related death (26). As a result, albuminuria may reflect a worse systematic or tumor burden and is related with increased risk of cancer death in RCC patients, associated with worse survival and more aggressive disease (26).”]

Response: Thank you again for your notice. The following sentece was added in the text.

Comment: Kindly revise the reference list according to the recommendations of the Biomedicines Journal (e.g., the journal name should be in italics). Please follow the link below: https://www.mdpi.com/journal/biomedicines/instructions

Response: Thank you again for your notice. The reference list was revised as you proposed.

Reviewer 3 Report

Comments and Suggestions for Authors

Authors have addressed all of my comments/suggestions in their revised submission. 

Author Response

Authors have addressed all of my comments/suggestions in their revised submission. 

Thank you for your suggestions and the chance to improve our article.

Reviewer 4 Report

Comments and Suggestions for Authors

The authors have fulfilled the required comments, and no further comments are requested.

Author Response

The authors have fulfilled the required comments, and no further comments are requested.

Thank you for your suggestions and the chance to improve our article.